# Peer review of "HPV Vaccination in Immunosuppressed Patients with Established Skin Warts and Non-Melanoma Skin Cancer: A Single-Institutional Cohort Study"

_vaccines, 2023, doi:10.3390/vaccines11091490_

Round 1
Reviewer 1 Report
I thank the authors for their manuscript entitled “HPV Vaccination in immunosuppressed patients with established skin warts and non-melanoma skin cancer: A single-institutional cohort study”. Here are my comments:
- Please consider analysing other groups of immunocompromised patients who did not receive the alpha HPV vaccination or who received an alternative alpha HPV vaccination.
- Please describe the clinical outcome/histological data of the non cancerous lesion for the different major dermatological interventions (biopsies, curettages, excisions). This is critical to evaluate the actual importance in reduction of interventions.
- Please better describe the time to event analysis and the hazard ratio
- Please comment more extensively table 4 and the outcome of the regression analysis
Please check for grammatical errors
Author Response
Dear Editor
We are pleased to submit our revised manuscript entitled “HPV Vaccination in immuno-suppressed patients with established skin warts and non-melanoma skin cancer: A single-institutional cohort study”. We appreciate the positive feedback from the editor and the constructive comments of the reviewers.
Following their suggestions, we have revised our manuscript. All changes are highlighted using the track changes mode. Please find our answer point by point.
- Reviewer 1, Comment 1: Please consider analysing other groups of immunocompro-mised patients who did not receive the alpha HPV vaccination or who received an alternative alpha HPV vaccination.
- Thank you for the comment. Our study is an observational study of patients who have received alpha-HPV vaccination. A comparison with another group would be a deviation from the study design. Also, alternative HPV vaccines are currently not approved in Switzerland.
- Reviewer 1, Comment 2: Please describe the clinical outcome/histological data of the non cancerous lesion for the different major dermatological interventions (biopsies, curettages, excisions). This is critical to evaluate the actual importance in reduction of interventions.
- Most non-cancerous lesions, mostly skin warts were not routinely biopsied as this was not a medical indication. Likewise, we do not have standardized photographic documentation. We can therefore only measure the effect with the number of therapy sessions, types of therapy and number of visits.
- Reviewer 1, Comment 3: Please better describe the time to event analysis and the hazard ratio.
- We have invested in clarification of the statistical methodology in the methods sections, including in the description of the PWP model.
- Reviewer 1, Comment 4: Please comment more extensively table 4 and the outcome of the regression analysis.
- We revised methodology, results and comments of the GLM model in the results and discussion section.
All of the authors have read and approved the paper. It has not been published previously nor is it being considered for publication by any other peer-reviewed journal. We thank for the reviewer his helpful comments and hope that our revised manuscript will be accepted.
Sincerely yours,
Simon Bossart, M.D., Corresponding-Author
Reviewer 2 Report
The authors presented an interesting manuscript referred to the effect of HPV vaccination on decrease of dermatologic interventions. Despite significant findings, there are important issues needed to be addressed in order to procced for publication.
First of all, a more detailed and analytic presentation of methods is necessary. For example, inclusion and exclusion criteria should be defined clearly (how many patients were excluded, reasons for excluded), time period for search in the registry to identify eligible patients. Also, s81 please check the type of vaccine received.
Also, the desctiption of exposure and possible predictors would be informative. There are some variables in table 1 not defined (i.e. tumor, dermatological). In addition to, there are 8 patients who did not received immunosuppressive agents. Are they considered immunosuppressed, and if yes, please explain the reasons for not receiving and the type of disease diagnosed.
Endpoints are defined clearly.
Statistical analysis: s99-102 needs amendment. It would be interesting also to examine the effect of Covid-19 as an independent predictors in a univariate analysis to see if it would be signicanlty related to the decrease of dermatologic interventions (as supp. Material for example). I believe would enhance even more the validity of the results.
Another point is that the sum of major/minor interventions presented in the table below figure (please add table number and title), do not sum up with the results presented in table 2. In table 2 are they presented the amount of dermatologic interventions or the number of patients received the interventions. Also, the decline in interventions are based on the amount of interventions or the amount of intervention per patient per year. Please define and also a description of the above in statistical analysis would be helpful.
The analysis with GLM is particularly interesting. Some points, which in my view, will make the results more robust. Was that analysis include only responders , or include the difference in major interventions, including also “responders” and “non-responders”? If no, it would interesting if an additional for the above and also for overall interventions and major interventions (as supplementary?) to be conducted. Moreover, how heavy pretreatment intervention is defined and why p-value from the variable in not significant when the CI is?
Discussion: limitation paragraph is adequately presented, but please place it before conclusion. Also, a future perspective would be interesting.
In tables and figures and text, please define abbreviations, explain all the metrices (i.e. age median, IQR) and describe legends in a more explanatory way. It would be much easier for the reader to understand the key points and the manuscript.
Refs are adequately presented.
Minor editing needed.
Author Response
Dear Editor
We are pleased to submit our revised manuscript entitled “HPV Vaccination in immuno-suppressed patients with established skin warts and non-melanoma skin cancer: A single-institutional cohort study”. We appreciate the positive feedback from the editor and the constructive comments of the reviewers.
Following their suggestions, we have revised our manuscript. All changes are highlighted using the track changes mode. Please find our answer point by point.
- Reviewer 2, Comment 1: First of all, a more detailed and analytic presentation of methods is necessary. For example, inclusion and exclusion criteria should be defined clearly (how many patients were excluded, reasons for excluded), time period for search in the registry to identify eligible patients. Also, s81 please check the type of vaccine received.
- As descriebed in the methods, patients with vaccination between December 5th, 2018 and January 1st, 2022 were analysed. We further clarified inclusion and exclusion criteria for the study in more details in the manuscript.
- Reviewer 2, Comment 2: Also, the descption of exposure and possible predictors would be informative. There are some variables in table 1 not defined (i.e. tumor, dermatological).
- We do not understand in detail the question and ask the reviewer to reformulate. Then, we are happy to perform additional analysis.
- Reviewer 2, Comment 3: In addition to, there are 8 patients who did not received immunosuppressive agents. Are they considered immunosuppressed, and if yes, please explain the reasons for not receiving and the type of disease diagnosed.
- As seen in Table 1, 8 patients were not immunosuppressed and did not receive immunosuppressive therapy. Yet all patients were suffering from hematological and/or solid tumors and received chemotherapy in the past. Therefore, the patients are considered immunosuppressed and at increased risk for non-melanoma skin cancer. This was clearly listed.
- Reviewer 2, Comment 4: Statistical analysis: s99-102 needs amendment. It would be interesting also to examine the effect of Covid-19 as an independent predictors in a univariate analysis to see if it would be signicanlty related to the decrease of dermatologic interventions (as supp. Material for example). I believe would enhance even more the validity of the results.
- We indeed included now all the independent predictors used in the model tables for the Cox-proportional hazard model of major dermatological interventions. National COVID Case load had no effect: HR 1.01 (CI: 0.99-1.03, p=0.5)
|
crude model |
intermediate model |
full model |
||||||||||
|
|
HR1 |
95% CI1 |
p-value |
HR1 |
95% CI1 |
p-value |
HR1 |
95% CI1 |
p-value |
|
||
|
Gardasil |
0.27 |
0.14, 0.51 |
<0.001 |
0.21 |
0.11, 0.41 |
<0.001 |
0.2 |
0.10, 0.41 |
<0.001 |
|
||
|
Gender [male] |
|
|
|
0.5 |
0.28, 0.88 |
0.017 |
0.5 |
0.28, 0.88 |
0.017 |
|
||
|
Age at vaccionation (per year) |
|
|
|
0.99 |
0.98, 1.00 |
0.13 |
0.99 |
0.98, 1.00 |
0.13 |
|
||
|
Indication [TPL] |
|
|
|
0.6 |
0.31, 1.19 |
0.15 |
0.6 |
0.31, 1.16 |
0.13 |
|
||
|
Skin lesion [SCC Tumor] |
|
|
|
3.69 |
2.31, 5.89 |
<0.001 |
3.68 |
2.31, 5.86 |
<0.001 |
|
||
|
national COVID Cases [per 1000] |
|
|
|
|
|
|
1.01 |
0.99, 1.03 |
0.5 |
|
||
- Reviewer 2, Comment 5: Another point is that the sum of major/minor interventions presented in the table below figure (please add table number and title), do not sum up with the results presented in table 2. In table 2 are they presented the amount of dermatologic interventions or the number of patients received the interventions. Also, the decline in interventions are based on the amount of interventions or the amount of intervention per patient per year. Please define and also a description of the above in statistical analysis would be helpful.
- Table 2 shows the number of visits with no, (only) minor or major interventions. A visit with major intervention is the primary outcome of the study. Meanwhile supplementary table 2 (no labeled) shows overall interventions through out the study. Patients may have received multiple interventions (multiple minors, multiple majors or mixed) at one visit. Therefore, the numbers do not add up. We clarified these details in the methods section
- Reviewer 2, Comment 6: The analysis with GLM is particularly interesting. Some points, which in my view, will make the results more robust. Was that analysis include only responders , or include the difference in major interventions, including also “responders” and “non-responders”? If no, it would interesting if an additional for the above and also for overall interventions and major interventions (as supplementary?) to be conducted. Moreover, how heavy pretreatment intervention is defined and why p-value from the variable in not significant when the CI is?
- We indeed expanded this analysis and performed logistic regression analysis for responders in respect of minor interventions, major interventions and overall interventions. In trend, patients with a high pre-vaccination intervention burden (top quartile of patients in respect of annual major interventions before first vaccine) and TPL patients were more likely to be responders, although data was only significant for high interventional burden an major interventions.
|
|
responder major interventions |
responder minor interventions |
responder overall interventions |
|
|||||||
|
OR |
95% CI1 |
p-value |
OR |
95% CI1 |
p-value |
OR |
95% CI1 |
p-value |
|
||
|
high intervention burden |
1.53 |
1.08, 2.18 |
0.024 |
0.76 |
0.54, 1.08 |
0.14 |
1.19 |
0.94, 1.53 |
0.2 |
|
|
|
age at vacc (years) |
1 |
1.0, 1.01 |
0.4 |
1 |
0.99, 1.01 |
>0.9 |
1 |
0.99, 1.00 |
0.7 |
|
|
|
no of IS (per 1) |
0.95 |
0.75, 1.22 |
0.7 |
1.17 |
0.93, 1.48 |
0.2 |
1 |
0.84, 1.19 |
>0.9 |
|
|
|
indication (TPL) |
1.37 |
0.76, 2.47 |
0.3 |
0.7 |
0.40, 1.23 |
0.2 |
1.23 |
0.81, 1.86 |
0.3 |
|
|
|
|
|||||||||||
- Reviewer 2, Comment 7: Discussion: limitation paragraph is adequately presented, but please place it before conclusion. Also, a future perspective would be interesting.
- We changed the order of the paragraphs in the discussion section. Already in the first version, we state that these findings should stimulate randomised interventional trials for primary (before IS) and secondary (after start IS) prophylaxic Gardasil vaccination in respect of major dermatological interventions.
- Reviewer 2, Comment 8: In tables and figures and text, please define abbreviations, explain all the metrices (i.e. age median, IQR) and describe legends in a more explanatory way. It would be much easier for the reader to understand the key points and the manuscript.
- We adapted the manuscript accordingly
All of the authors have read and approved the paper. It has not been published previously nor is it being considered for publication by any other peer-reviewed journal. We thank for the reviewer his helpful comments and hope that our revised manuscript will be accepted.
Sincerely yours,
Simon Bossart, M.D., Corresponding-Author

Reviewer 3 Report
The present article reports about a retrospective study to investigate the therapeutic potential of alpha-HPV vaccination for immunosuppressed patients with cutaneous SCC and its precursors. The authors have tried to determine the effect of Gardasil 9 as dermatologic intervention in immunosuppressed patients.
Major Comments:
1. HPV vaccination works only if administered before sexual debut. How is current intervention relevant with respect to HPV associated established NMSC & cSCC?
2. There is no data about prevalence of HPV in the current cohort and whether the dermatological manifestation of NMSC & cSCC is associated with the same. Kindly elaborate.
3. The data doesn’t represent female population.
4. There is no significant difference between data for patient interventions with respect to the administration of Gardasil 9 vaccine.
5. Due to insufficient data about HPV infection status among the population and antibody generation after the administration of vaccine, there is no clear idea if vaccine administration is contributing to decrease in surgical interventions.
6. There is no data on the histopathology and HPV status of the warts. As a result, it is not clear if vaccine is contributing to the reduction of the same and eventually reduction in surgical procedures.
7. The cohort has very low number of patients studied resulting in rather insignificant outcome from the study. This is one of the major limitations of the study.
Minor Comments:
1. Line 12 – Replace carcinom with carcinoma
2. Line 14 – Replace therapeutically with therapeutic.
3. Line 48 – Kindly correct the word orogenital pharynx. Replace it with anogenital and oropharyngeal.
Minor typological and grammatical errors, throughout the text, need to be addressed carefully.
Author Response
Dear Editor
We are pleased to submit our revised manuscript entitled “HPV Vaccination in immuno-suppressed patients with established skin warts and non-melanoma skin cancer: A single-institutional cohort study”. We appreciate the positive feedback from the editor and the constructive comments of the reviewers.
Following their suggestions, we have revised our manuscript. All changes are highlighted using the track changes mode. Please find our answer point by point.
- Reviewer 3, Comment 1: HPV vaccination works only if administered before sexual debut. How is current intervention relevant with respect to HPV associated established NMSC & cSCC?
- We discussed this in the introduction in lines 86-92. There is evidence that HPV vaccination against alpha-genera has also resulted in regression of cutaneous warts and cutaneous SCCs. That is why we did this study.
- Reviewer 3, Comment 2: There is no data about prevalence of HPV in the current cohort and whether the dermatological manifestation of NMSC & cSCC is associated with the same. Kindly elaborate.
- This is correct. We have not done routine HPV analysis and calculation of exact prevalence, as we have not performed conventional HPV typing for beta strains commercially in hospitals in Switzerland. Nevertheless, it is a fact that especially immunosuppressed and transplanted patients are infected with HPV in 65-90%. We have listed this in the Itnroduction on line 67-70. It is therefore reasonable to assume that the majority of our cohort is infected with HPV.
- Reviewer 3, Comment 3: The data doesn’t represent female population.
- 24% of patients were female, see Table 1 Patient Characteristics.
- Reviewer 3, Comment 4: There is no significant difference between data for patient interventions with respect to the administration of Gardasil 9 vaccine.
- Administration of Gardasil 9 vaccine was performed in all patients according to the manufacturer's instructions in the vaccination schedule month 0, 2 and 6.
- Reviewer 3, Comment 5: Due to insufficient data about HPV infection status among the population and antibody generation after the administration of vaccine, there is no clear idea if vaccine administration is contributing to decrease in surgical interventions.
- That is absolutely correct. We have stated this in the Limitation Discussion p. 213-222. It needs large-scale prospective RCTs for definitive confirmation.
- Reviewer 3, Comment 6: There is no data on the histopathology and HPV status of the warts. As a result, it is not clear if vaccine is contributing to the reduction of the same and eventually reduction in surgical procedures.
- This is correct and also so listed in Limition P. 213-222. Histopathology and HPV typing of skin warts are lacking and unfortunately were not routinely collected. Skin warts are diagnosed clinically by dermoscopic examination. Our study has a retrospective design. Retrospective collection of these data is not possible.
- Reviewer 3, Comment 7: The cohort has very low number of patients studied resulting in rather insignificant outcome from the study. This is one of the major limitations of the study.
- This is true and has been clearly described in the limitation on page 213-215: "..the rather low number of patients does not allow to exclude all co-founders with may influence study results (patient no-shows, drop-outs)."
- Reviewer 3, Comment 8: Line 12 – Replace carcinom with carcinoma.
- We have changed this accordingly.
- Reviewer 3, Comment 9: Line 14 – Replace therapeutically with therapeutic.
- We have changed this accordingly.
- Reviewer 3, Comment 10: Line 48 – Kindly correct the word orogenital pharynx. Replace it with anogenital and oropharyngeal.
- We have changed this accordingly.
All of the authors have read and approved the paper. It has not been published previously nor is it being considered for publication by any other peer-reviewed journal. We thank for the reviewer his helpful comments and hope that our revised manuscript will be accepted.
Sincerely yours,
Simon Bossart, M.D., Corresponding-Author

Round 2
Reviewer 1 Report
I thank the authors for the changes made in the manuscript. The manuscript is now ready to be published as an observational study of patients who have received alpha-HPV vaccination.
However, histopathological data, info on HPV infections and seroconversion as well as comparison with other treatments or untreated patients are necessary for a complete understanding of the actual contribution of the vaccine. Some of these limitations are already listed in the final discussion. Please, mention all of them.
Author Response
Dear Editor
We are pleased to submit our revised manuscript entitled “HPV Vaccination in immuno-suppressed patients with established skin warts and non-melanoma skin cancer: A single-institutional cohort study”. We appreciate the positive feedback from the editor and the constructive comments of the reviewers.
Following their suggestions, we have revised our manuscript. All changes are highlighted using the track changes mode. Please find our answer point by point.
- Reviewer 1, Comment: I thank the authors for the changes made in the manuscript. The manuscript is now ready to be published as an observational study of patients who have received alpha-HPV vaccination. However, histopathological data, info on HPV infections and seroconversion as well as comparison with other treatments or untreated patients are necessary for a complete understanding of the actual contribution of the vaccine. Some of these limitations are already listed in the final discussion. Please, mention all of them.
- Thank you for the comment. We have added the missing limitation under Limitation in Discussion.
- Editor Comment: We noticed that the current manuscript has 3604 words in the main text. We suggested you enrich the main body above 4000 words during your revision.
- Thank you for the comment. We have now enriched the text to over 4000 words (references excluded) with additions in the Introduction section.
All of the authors have read and approved the paper. It has not been published previously nor is it being considered for publication by any other peer-reviewed journal. We thank for the reviewer his helpful comments and hope that our revised manuscript will be accepted.
Sincerely yours,
Simon Bossart, M.D., Corresponding-Author
Reviewer 2 Report
The manuscript has been significantly improved after changes and important issues have been addressed adequately by the authors.
One minor comment : please define in methods which were the type of cancer and dermatologic issues presented in table 1.
Minor editing needed.
Author Response
Dear Editor
We are pleased to submit our revised manuscript entitled “HPV Vaccination in immuno-suppressed patients with established skin warts and non-melanoma skin cancer: A single-institutional cohort study”. We appreciate the positive feedback from the editor and the constructive comments of the reviewers.
Following their suggestions, we have revised our manuscript. All changes are highlighted using the track changes mode. Please find our answer point by point.
- Reviewer 2, Comment: The manuscript has been significantly improved after changes and important issues have been addressed adequately by the authors. One minor comment: please define in methods which were the type of cancer and dermatologic issues presented in table 1.
- Thank you for your comment. We have clarified and corrected this both in Table 1 and in the Results section on page 7.
- Editor Comment: We noticed that the current manuscript has 3604 words in the main text. We suggested you enrich the main body above 4000 words during your revision.
- Thank you for the comment. We have now enriched the text to over 4000 words (references excluded) with additions in the Introduction section.
All of the authors have read and approved the paper. It has not been published previously nor is it being considered for publication by any other peer-reviewed journal. We thank for the reviewer his helpful comments and hope that our revised manuscript will be accepted.
Sincerely yours,
Simon Bossart, M.D., Corresponding-Author
Reviewer 3 Report
Thank you for agreeing with most of the raised concerns and mentioning them in the manuscript as limitations. Addressing them with proper scientific support might enhance the novelty of the study.
Minor typos are still there which may need to correct during the proofread.
Author Response
Dear Editor
We are pleased to submit our revised manuscript entitled “HPV Vaccination in immuno-suppressed patients with established skin warts and non-melanoma skin cancer: A single-institutional cohort study”. We appreciate the positive feedback from the editor and the constructive comments of the reviewers.
Following their suggestions, we have revised our manuscript. All changes are highlighted using the track changes mode. Please find our answer point by point.
- Reviewer 3, Comment: Thank you for agreeing with most of the raised concerns and mentioning them in the manuscript as limitations. Addressing them with proper scientific support might enhance the novelty of the study. Minor typos are still there which may need to correct during the proofread.
- Thank you for the comment. We have corrected typos.
- Editor Comment: We noticed that the current manuscript has 3604 words in the main text. We suggested you enrich the main body above 4000 words during your revision.
- Thank you for the comment. We have now enriched the text to over 4000 words (references excluded) with additions in the Introduction section.
All of the authors have read and approved the paper. It has not been published previously nor is it being considered for publication by any other peer-reviewed journal. We thank for the reviewer his helpful comments and hope that our revised manuscript will be accepted.
Sincerely yours,
Simon Bossart, M.D., Corresponding-Author